# The Association between the *ALDH2* rs671 Polymorphism and Athletic Performance in Japanese Power and Strength Athletes

**DOI:** 10.3390/genes13101735

**Published:** 2022-09-27

**Authors:** Aoto Saito, Mika Saito, Kathleen Y. de Almeida, Hiroki Homma, Minoru Deguchi, Ayumu Kozuma, Naoyuki Kobatake, Takanobu Okamoto, Koichi Nakazato, Naoki Kikuchi

**Affiliations:** 1Graduate School of Health and Sport Science, Nippon Sport Science University, Tokyo 158-8508, Japan; 2Faculty of Sport Science, Nippon Sport Science University, Tokyo 158-8508, Japan; 3Faculty of Medical Science, Nippon Sport Science University, Tokyo 158-8508, Japan

**Keywords:** polymorphism, genotype, athletic performance, power/strength athletes, mitochondrial aldehyde dehydrogenase 2

## Abstract

The rs671 polymorphism is associated with the enzymatic activity of aldehyde dehydrogenase 2 (ALDH2), which is weakened by the A allele in East Asians. We recently reported the association of this polymorphism with the athletic status in athletic cohorts and the muscle strength of non-athletic cohorts. Therefore, we hypothesized the association of *ALDH2* rs671 polymorphism with the performance in power/strength athletes. We aimed to clarify the relationship between the *ALDH2* rs671 polymorphism and performance in power/strength athletes. Participants comprising 253 power/strength athletes (167 men and 86 women) and 721 healthy controls (303 men and 418 women) were investigated. The power/strength athletes were divided into classic powerlifting (*n* = 84) and weightlifting (*n* = 169). No differences in the genotypes and allele frequencies of the *ALDH2* rs671 polymorphism and an association between performance and the *ALDH2* rs671 genotype were observed in weightlifters. However, the relative values per body weight of the total record were lower in powerlifters with the GA + AA genotype than those with the GG genotype (7.1 ± 1.2 vs. 7.8 ± 1.0; *p* = 0.010, partial η^2^ = 0.08). Our results collectively indicate a role of the *ALDH2* rs671 polymorphism in strength performance in powerlifters.

## 1. Introduction

Various studies have reported that skeletal muscle hypertrophy, hyperplasia, the predominance of fast-twitch muscle fibers, improved neurological adaptation, and high glycolytic capacity are the main factors affecting the performance of power/strength athletes [1,2,3]. Moreover, power/strength athletes represent entirely different transcriptomic, biochemical, anthropometric, physiological, biomechanical, and other characteristics as compared with endurance athletes and non-athletic groups [4,5,6,7]. These differences are attributed to environmental (e.g., training and nutrition) and genetic factors. Some studies have demonstrated a strong heritability of power and strength-related characteristics, with genetic factors accounting for 30–85% of the variation in isometric, isotonic, isokinetic strength, jumping ability, and other muscle strength phenotypes [8,9]. Furthermore, a previous study has indicated that at least 69 genetic markers, such as single nucleotide polymorphisms, are associated with power athlete status [10]. Moreover, the genetic characteristics related to power/strength athlete status differ according to ethnicity [11].

The *ALDH2* rs671 polymorphism, a characteristic of East Asians, is associated with the weakened enzymatic activity of aldehyde dehydrogenase 2 (ALDH2). Approximately 30–50% of East Asians carry the *ALDH2* single nucleotide polymorphism, where heterozygosity (GA genotype) and homozygosity (AA genotype) lead to loss of function and adversely affect acetaldehyde metabolism, causing the alcohol flush reaction [12], which is the result of an accumulation of acetaldehyde and is caused by an aldehyde dehydrogenase 2 deficiency. Previous studies have confirmed that *ALDH2* rs671 polymorphism is associated with body mass index (BMI) [13], hypertension [14], and cancer [15], presumably due to the reduced capacity of inactivated ALDH2 to attenuate oxidative stress.

Recently, Wakabayashi et al. [16] reported that ALDH2 is expressed in skeletal muscle mitochondria and that ALDH2-deficient mice exhibit increased mitochondrial reactive oxidative stress (ROS) emission. Therefore, we hypothesized that the *ALDH2* rs671 polymorphism might have deleterious effects on human skeletal muscles. 

Furthermore, we recently reported that the AA genotype and A allele frequency were lower in athletes, especially mixed athletes (e.g., American football, baseball, basketball, volleyball, handball, rugby, soccer, and others). In addition, comparing the muscle strength of those with the *ALDH2* rs671 polymorphism in non-athletic cohorts revealed that the AA genotype-harboring individuals have a lower grip strength than the G allele-harboring ones [17]. Therefore, we suggested that the AA genotype and A allele of the *ALDH2* rs671 polymorphism are associated with a reduced athletic capacity and poorer muscle phenotype.

It has been reported that high-intensity anaerobic exercise induces increased oxidative stress [18]. Moreover, studies have reported increased oxidative stress in power/strength athletes after a strength-training period, suggesting that high-intensity strength training may adversely affect their strength performance due to exercise-induced high oxidative stress [19,20]. In addition, Ahmetov et al. [21] reported that antioxidant-related genetic markers are associated with power/strength athletes. Therefore, we hypothesized that the *ALDH2* rs671 polymorphism may affect the phenotype of competitive performance in power/strength athletes. Thus, we aimed to clarify the relationship between the *ALDH2* rs671 polymorphism and the performance of power/strength athletes.

## 2. Materials and Methods

### 2.1. Participants

This study enrolled 974 Japanese individuals, comprising 253 power/strength athletes (167 men and 86 women) and 721 healthy controls (303 men and 418 women). The power/strength athletes were divided into two groups: classic powerlifting (*n* = 84) and weightlifting (*n* = 169). Additionally, powerlifters and weightlifters were divided into the following categories: international, national, and regional, based on competition results. Power/strength athletes were included in this study if they had participated in official competitions (classic powerlifting or weightlifting), with their official records in these competitions verified in the questionnaire. Furthermore, we included a non-athletic cohort from Tokyo and surrounding areas as controls. All participants provided written informed consent for their participation in the study, and the protocol was approved by the Nippon Sport Science University ethics committee (020-G03) and conducted by strictly following the tenets of the Declaration of Helsinki.

### 2.2. Athletic Performance

In powerlifting, the total lifting weight of squats, bench presses, and deadlifts is considered in each weight class. In weightlifting, the total lifting weight of snatch and clean and jerk are considered in each weight class. Thus, according to the questionnaire, powerlifting, and weightlifting performances were decided by considering answers with the best total record and its breakdown in each subject in the official powerlifting or weightlifting competition. Moreover, relative values per body weight of the record (squats, bench presses, deadlifts, and total record) in powerlifters and the relative values per body weight of the record (snatch, clean and jerk, and total record) in weightlifters were calculated by dividing each record by the body weight at the time when the best total records were achieved.

### 2.3. Genotyping

Total DNA was extracted from saliva samples using an Oragene-DNA Kit (DNA Genotek, Ontario, Canada). Saliva from the participants was collected in the laboratory using a kit. Alternatively, the kit was mailed to the participant’s home, and the saliva samples were collected and returned to the laboratory. The *ALDH2* rs671 polymorphism was identified using the TaqMan SNP Genotyping Assay (C__590093_1_), which was conducted using the TaqMan Real-Time PCR System (Applied Biosystems, Foster City, CA, USA). 

### 2.4. Statistical Analyses

All statistical analyses were performed using the SPSS Statistics software version 27 (IBM Japan, Tokyo, Japan). The Pearson’s chi-square test was used to confirm the Hardy–Weinberg equilibrium in the observed genotype frequencies and to compare the *ALDH2* rs671 alleles and genotype frequencies between power/strength athletes and controls. Furthermore, the Pearson’s chi-square test was used to compare the sex and athletic status among each genotype.

The characteristics and performance of the powerlifters (age, height, body mass, powerlifting experience, squat, bench press, deadlift, and total record) and the weightlifters (age, height, body mass, weightlifting experience, snatch, clean and jerk, and total record) among each genotype were analyzed using a one-way analysis of variance and an analysis of covariance adjusted by sex (because of the significant difference in absolute and relative performance between men and women). The magnitude of the effect size was also measured and was reported using partial eta squared (partial η^2^). *p* values < 0.05 were considered statistically significant.

## 3. Results

The *ALDH2* rs671 genotype frequencies were in a Hardy–Weinberg equilibrium among power/strength athletes and controls. The genotype frequencies are shown in Table 1. We observed no differences in genotype frequencies between power/strength athletes, powerlifters, weightlifters, and controls. These results were similar to those reported for other Japanese populations [17].

The powerlifter characteristics (sex, athletic status, age, height, body mass and powerlifting experience) and performance (squat, bench press, deadlift, and total record) of the *ALDH2* rs671 genotype are shown in Table 2. None of the participant characteristics (sex, athletic status, age, height, body mass, and powerlifting experience) differed among the *ALDH2* rs671 genotypes in the powerlifters. A difference in the performance of relative values per body weight of the total record in powerlifters was observed between the three *ALDH2* genotypes (*p* = 0.031) (Bold in Table.2).

Moreover, the relative values per body weight of the total record were lower in powerlifters with the GA + AA genotype than those with the GG genotype (relative values per body weight of the total record: 7.1 ± 1.2 vs. 7.8 ± 1.0; *p* = 0.010, partial η^2^ = 0.08, total record: 555.3 ± 138.5 vs. 592.7 ± 121.7; *p* = 0.455, partial η^2^ = 0.02, Figure 1). 

The weightlifter characteristics (sex, athletic status, age, height, body mass, and weightlifting experience) and performance (snatch, clean and jerk, and total record) related to the *ALDH2* rs671 genotype are shown in Table 3. None of the participant characteristics (sex, athletic status, age, height, body mass, and weightlifting experience) differed among the *ALDH2* rs671 genotypes in the weightlifters. 

No difference in performance was observed between the weightlifters harboring the three *ALDH2* genotypes, as seen in Table 3. Moreover, no significant difference was observed between weightlifters with the GA + AA genotype and GG genotype (relative values per bo dyweight of the total record: 3.2 ± 0.6 vs. 3.2 ± 0.6; *p* = 0.932, partial η^2^ = 0.001, total record: 222.6 ± 57.0 vs. 232.1 ± 63.2; *p* = 0.371, partial η^2^ = 0.006, Figure 2). 

## 4. Discussion

This is the first study to investigate the association between the *ALDH2* rs671 polymorphism and athletic performance in power/strength athletes. In our recent study, we investigated the relationship between the *ALDH2* rs671 polymorphism and athletic status in a Japanese athletic cohort, including sprint/power (*n* = 822), mixed (*n* = 701), and endurance athletes (*n* = 191). The AA genotype and A allele frequencies of the *ALDH2* rs671 polymorphism were lower in all athletes (*n* = 1714) than controls, especially in mixed athletes. However, no difference in genotype frequencies between sprint/power athletes and controls was observed [17]. Furthermore, this study failed to confirm a relationship between the *ALDH2* rs671 polymorphism and athletic status in power/strength athletes. 

We assessed the relationship between the *ALDH2* rs671 polymorphism and powerlifting performance (squat, bench press, deadlift, and total record) and weightlifting performance (snatch, clean and jerk, and total record). We observed that powerlifters with the GG genotype, who were previously reported to have athlete-related variants [17], have higher performance than powerlifters with the GA and AA genotypes (relative values per bodyweight of the total record: *p* = 0.010, partial η^2^ = 0.08). However, no association between the genotype and performance was observed in weightlifters. 

Our previous study reported that the AA genotype had a lower maximal voluntary contraction (MVC) compared to those with the G allele in non-athletic cohorts [17]. Therefore, the *ALDH2* rs671 polymorphism may affect muscle functions such as MVC. In the present study, our results indicated a relationship between the *ALDH2* rs671 polymorphism and powerlifting performance; those with the A allele had a lower performance than those with the GG genotype. Powerlifting consists of squats, bench presses, and deadlifts, requiring extreme muscle strength with simple movements analogous to MVC. In contrast, weightlifting performance (snatch and clean and jerk) needs muscle strength, power, and specific techniques. Therefore, we could replicate the association between muscle function (powerlifting performance) and the *ALDH2* rs671 polymorphism in athletes.

ALDH2 is expressed in skeletal muscle mitochondria, and ALDH2-deficient mice have high skeletal muscle mitochondrial ROS generation [16], suggesting that ALDH2 inactivation impairs mitochondrial function and skeletal muscle. This observation is attributed to the fact that mitochondrial functions are strongly associated with skeletal muscle health [22]. Consistent with these findings, Kobayashi et al. [23] showed a smaller muscle fiber diameter in transgenic mice overexpressing ALDH2*2 encoded by the *ALDH2* A allele compared to the wild-type mice. In addition, a recent study by Kasai and collaborators [24] showed that aged mice with an ALDH2 deficiency showed a reduced cross-sectional area of skeletal muscle fibers of the soleus. These studies indicated that ALDH2 deficiency is a consequence of muscle atrophy. There are grounds to believe that a similar situation might also arise in humans. Our recent study compared the MVC of subjects with the *ALDH2* rs671 polymorphism in the non-athletic cohorts, which revealed that the AA genotype-harboring individuals had a lower MVC than the G allele-harboring ones [17]. Therefore, we suggest that the A allele of the *ALDH2* rs671 polymorphism adversely affects the muscle strength phenotype by impairing muscle configuration and function in competitive human athletes, based on observations in mice and those made in this study. 

We attributed the different performance results of powerlifters and weightlifters to their different muscle strength requirements. As previously mentioned, powerlifting involves the use of extreme muscle strength with simple movements analogous to MVC, while weightlifting requires fast and complex movements. A recent study reported that powerlifters and weightlifters possessed a few different DNA polymorphisms associated with power and strength status, respectively [25], suggesting that powerlifters and weightlifters have different genetic traits associated with each performance. Moreover, in a recent meta-analysis, Chung et al. [26] have shown that strength and power phenotypes respond differently and play different roles according to genetic variability. Therefore, the results of this study indicate the importance of examining the association between phenotypes and polymorphisms by differentiating between power-oriented (e.g., weightlifters) and strength-oriented (e.g., powerlifters) athletes. In addition, all powerlifters and weightlifters who participated in this study could not achieve their best performance (e.g., due to different athletic experiences). Moreover, regarding age, weightlifters tended to be younger than powerlifters (mean age of weightlifters was 20.9 ± 3.7, while that of powerlifters was 29.2 ± 9.2). It has been shown that aging is related to increased oxidative stress levels [27]. However, the powerlifter group included younger individuals; therefore, this effect might not significantly affect the association between their athletic performance and *ALDH2* rs671 polymorphism. Moreover, there was also a difference between powerlifters and weightlifters regarding the ratio of men to women (weightlifters, men: 96 (57%): women: 73 (43%), while powerlifters: 71 (85%): women: 13 (15%). Although we have considered sex as a variable in their performance, differences in testosterone levels strongly affected by sex are also important in power/strength athletes [28]. As a result, those characteristics may have affected the association between performance and the *ALDH2* rs671 polymorphism in powerlifters and weightlifters. However, how *ALDH2* rs671 polymorphisms influence the differences in competitive performance in powerlifters and weightlifters is unknown and requires further study.

ALDH2 detoxifies the acetaldehyde produced from ethanol, suggesting a strong association of *ALDH2* rs671 polymorphisms with alcohol consumption [29]. Alcohol consumption could lead to mitochondrial dysfunction and increased oxidative stress [30], suggesting that it could potentially affect muscle functions [31]. Therefore, alcohol consumption levels may affect the relationship between *ALDH2* rs671 polymorphisms and muscle functions. Further research is necessary to reveal the effect of alcohol consumption and *ALDH2* rs671 polymorphisms on muscle functions in athletes. 

This study has some limitations. We failed to consider factors, such as alcohol consumption habits and the daily training status of each athlete, which probably influence athletic conditions and performance. Therefore, longitudinal studies are necessary to explain the effect of the *ALDH2* rs671 polymorphism in response to aging, physical training, and alcohol consumption levels. In addition to the effects of this polymorphism, other genetic polymorphisms, such as *SOD2* rs4880, as well as environmental factors can probably interact and influence muscular functions. The application of high-throughput technologies in genome-wide association studies, such as next-generation whole genome and/or exome sequencing, is warranted to help uncover these multiple genetic effects. 

## 5. Conclusions

In conclusion, our results indicate that the *ALDH2* rs671 polymorphism is associated with strength performance in powerlifters. However, we were not able to observe the same association in weightlifters. Therefore, further research must aim to clarify the association between athletic performance and the *ALDH2* rs671 polymorphism in power/strength athletes. Furthermore, The ALDH2 rs671 genotype might provide useful information (e.g., anti-oxidative supplement and genotype-based customization for conditioning) for athletes, especially in strength related athletes and their strength and conditioning coaches and sports coaches.

## Figures and Tables

**Figure 1 genes-13-01735-f001:**
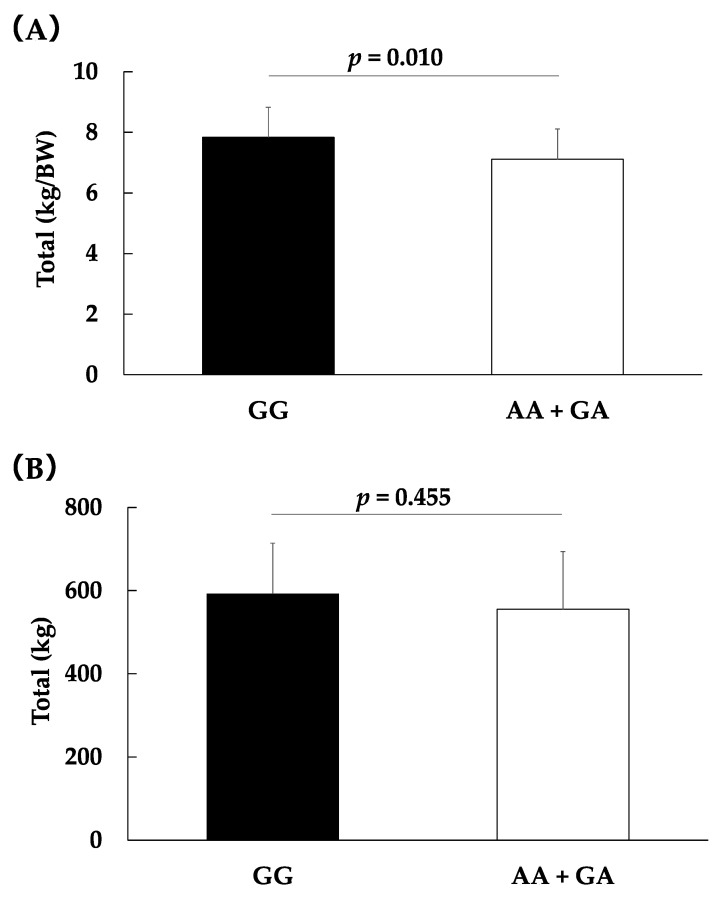
Associations between the *ALDH2* rs671 genotype and relative values of the total record per body weight (**A**) and the total record (**B**) in powerlifters. Data are shown as the means ± SD. The data were assessed by analysis of covariance with adjustments for sex as covariates.

**Figure 2 genes-13-01735-f002:**
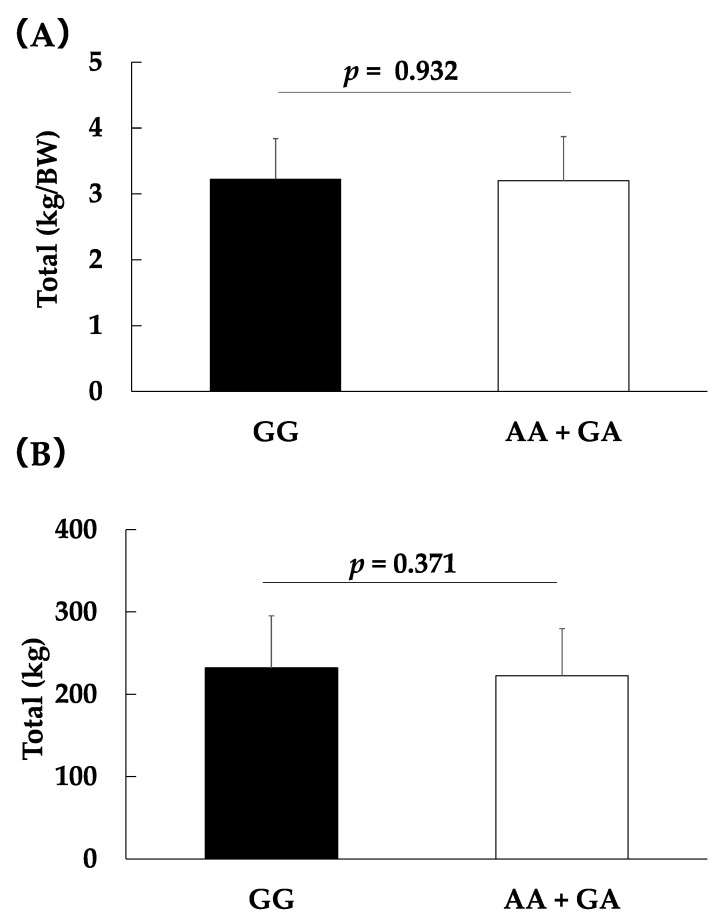
Associations between the *ALDH2* rs671 genotype and relative values of the total record per body weight (**A**) and the total record (**B**) in weightlifters. Data are shown as the means ± SD. The data were assessed by analysis of covariance with adjustments for sex as covariates.

**Table 1 genes-13-01735-t001:** The genotypes and allele frequencies of *ALDH2* rs671 polymorphism.in power/strength athletes and controls.

		Genotype			Allele		*p* Value			
	N	GG	GA	AA	G	A	Genotype	Dominant	Recessive	Allele
		*n* (%)	*n* (%)	*n* (%)	*n* (%)	*n* (%)				
All power/strength athletes	253	140 (56)	93 (38)	20 (6)	532 (75)	182 (25)	0.688	0.502	0.465	0.384
Powerlifters	84	50 (60)	28 (33)	6 (7)	128 (76)	40 (24)	0.929	0.866	0.785	0.879
Weightlifters	169	90 (53)	65 (39)	14 (8)	245 (73)	93 (27)	0.491	0.454	0.264	0.224
controls	721	418 (58)	255 (35)	48 (7)	1091 (76)	351 (24)				

**Table 2 genes-13-01735-t002:** Characteristics and performance of powerlifters with an *ALDH2* genotype.

	GG (*n* = 50)	GA (*n* = 28)	AA (*n* = 6)	GA + AA (*n* = 34)	*p* Value (Genotype)	ES (Genotype)	*p* Value (Recessive)	ES (Recessive)
Sex								
Men	*n* = 44 (88%)	*n* = 22 (79%)	*n* = 5 (83%)	*n* = 27 (79%)	0.541 *		0.285 *	
Women	*n* = 6 (12%)	*n* = 6 (21%)	*n* = 1 (17%)	*n* = 7 (21%)				
Athletic status								
International	*n* = 16 (32%)	*n* = 12 (43%)	*n* = 1 (17%)	*n* = 13 (38%)	0.158 *		0.112 *	
National	*n* = 25 (50%)	*n* = 16 (57%)	*n* = 4 (66%)	*n* = 20 (59%)				
Regional	*n* = 9 (18%)	*n* = 0 (0%)	*n* = 1 (17%)	*n* = 1 (3%)				
Age, yr	28.8 ± 7.7	29.9 ± 12.1	30.1 ± 5.3	30.0 ± 11.2	0.841	0.004	0.556	0.004
Height, cm	167.2 ± 7.1	166.6 ± 8.1	168.3 ± 10.2	166.9 ± 8.4	0.871	0.003	0.873	0.0003
Body mass, kg	76.2 ± 17.3	78.2 ± 21.1	82.9 ± 26.0	79.1 ± 21.7	0.691	0.009	0.503	0.005
Powerlifting experience, yr	4.9 ± 5.0	4.5 ± 5.2	5.7 ± 6.4	4.8 ± 5.4	0.862	0.004	0.890	0.0002
Squat, kg	211.9 ± 47.6	196.1 ± 54.1	216.2 ± 59.2	199.7 ± 54.7	0.553 ^†^	0.02	0.665 ^†^	0.01
Bench press, kg	145.1 ± 35.6	129.6 ± 38.1	159.5 ± 51.4	134.9 ± 41.6	**0.049** ^†^	0.05	0.554 ^†^	0.01
Deadlift, kg	236.4 ± 43.1	220.4 ± 50.6	225.8 ± 50.3	221.4 ± 49.9	0.620 ^†^	0.02	0.313 ^†^	0.02
Squat, kg/BW	2.7 ± 0.3	2.5 ± 0.4	2.6 ± 0.5	2.5 ± 0.4	**0.032** ^†^	0.09	**0.011** ^†^	0.08
Bench press, kg/BW	1.9 ± 0.3	1.6 ± 0.3	1.9 ± 0.5	1.7 ± 0.4	**0.025** ^†^	0.09	**0.047** ^†^	0.06
Deadlift, kg/BW	3.1 ± 0.4	2.8 ± 0.5	2.8 ± 0.5	2.8 ± 0.5	0.071^†^	0.07	**0.023** ^†^	0.07
Total, kg	592.7 ± 121.7	545.6 ± 137.6	600.8 ± 146.2	555.3 ± 138.5	0.363	0.03	0.455	0.02
Total, kg/BW	7.8 ± 1.0	7.0 ± 1.2	7.4 ± 1.3	7.1 ± 1.2	**0.031** ^†^	0.09	**0.011** ^†^	0.08

Data are shown as the means ± SD. * Pearson’s chi-square test. ^†^ ANCOVA adjusted by sex. ES: effect size.

**Table 3 genes-13-01735-t003:** Characteristics and performance of weightlifters with an *ALDH2* genotype.

	GG (*n* = 90)	GA (*n* = 65)	AA (*n* = 14)	GA + AA (*n* = 79)	*p* Value (Genotype)	ES (Genotype)	*p* Value (Recessive)	ES (Recessive)
Sex								
Men	*n* = 53 (59%)	*n* = 37 (57%)	*n* = 6 (43%)	*n* = 43 (54%)	0.530 *		0.559 *	
Women	*n* = 37 (41%)	*n* = 28 (43%)	*n* = 8 (57%)	*n* = 36 (46%)				
Athletic status								
International	*n* = 34 (38%)	*n* = 24 (37%)	*n* = 5 (36%)	*n* = 29 (37%)	0.999 *		0.973 *	
National	*n* = 39 (43%)	*n* = 28 (43%)	*n* = 6 (43%)	*n* = 34 (43%)				
Regional	*n* = 17 (19%)	*n* = 13 (20%)	*n* = 3 (21%)	*n* = 16 (20%)				
Age, yr	20.8 ± 3.1	21.0 ± 4.5	20.9 ± 4.0	21.0 ± 4.4	0.902	0.001	0.666	0.001
Height, cm	163.4 ± 8.5	161.5 ± 8.7	162.6 ± 9.2	161.7 ± 8.7	0.406	0.01	0.201	0.01
Body mass, kg	72.7 ± 20.0	71.4 ± 19.2	67.9 ± 14.4	70.8 ± 18.4	0.673	0.005	0.520	0.002
weightlifting experience, yr	5.7 ± 2.6	6.1 ± 3.7	5.9 ± 3.8	6.1 ± 3.6	0.695	0.004	0.411	0.004
Snatch, kg	103.6 ± 29.1	99.7 ± 27.0	94.6 ± 21.4	98.8 ± 26.0	0.570 ^†^	0.01	0.294 ^†^	0.008
Clean and Jerk, kg	128.4 ± 34.3	125.3 ± 32.4	116.2 ± 25.0	123.7 ± 31.3	0.726 ^†^	0.01	0.458 ^†^	0.005
Snatch, kg/BW	1.4 ± 0.2	1.4 ± 0.3	1.4 ± 0.2	1.4 ± 0.2	0.873 ^†^	0.002	0.753 ^†^	0.002
Clean and Jerk, kg/BW	1.7 ± 0.3	1.7 ± 0.3	1.7 ± 0.3	1.7 ± 0.3	0.995 ^†^	0.001	0.922 ^†^	0.0001
Total, kg	232.1 ± 63.2	225.1 ± 59.1	210.2 ± 25.0	222.6 ± 57.0	0.669 ^†^	0.01	0.371 ^†^	0.006
Total, kg/BW	3.2 ± 0.6	3.2 ± 0.6	3.1 ± 0.6	3.2 ± 0.6	0.928 ^†^	0.001	0.932 ^†^	0.001

Data are shown as the means ± SD. * Pearson’s chi-square test. † ANCOVA adjusted by sex. ES: effect size.

## Data Availability

The data presented in this study are available on request from the corresponding author.

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
