# Peer review of "The Association between the ALDH2 rs671 Polymorphism and Athletic Performance in Japanese Power and Strength Athletes"

_genes, 2022, doi:10.3390/genes13101735_

Round 1

Reviewer 1 Report

1. Introduction

A larger physiological argument must be explored to justify a possible effect of polymorphism on muscle oxidative attack. The physiological link to the construction of the hypothesis around the polymorphism is the main limitation of the study.

What is the genotypic frequency in other populations of GG, AA, and GA?

2. Method

Inform the number of the ethics committee.

Include the criteria for classifying the international, national and regional levels of athletes. I suggest that a comparison be made between the levels of the athletes (international, regional, and national).

It is unclear how the absolute and relative Total Record by body mass was calculated.

Inform if the best results were self-declared, analyzed in loco or consulted in an official record.

General

Avoid the terms: "significant difference", if it was described that there was a difference, it does not need to be redundant to say that it was "significant" or "statistical"

The rationale for ALDH2 revolves around oxidative stress. Consequently, in older people, oxidative damage over the years shows greater cumulative damage. The average age of the powerlifters group was almost 10 years older than the other group. What do you think about it?

Why wasn't 3 groups analyzed: GG, AA and GA? Just GG vs AA+GA? The sample n seems to me enough to have statistical power in the subdivision into 3 groups.

What are the limitations of the study?

As in the Introduction, the physiological mechanism that explains the influence of polymorphisms on mitochondrial and muscle oxidative stress needs to be better connected.

The endogenous antioxidant system is the most important in combating reactive species. The adaptation of the antioxidant system depends on oxidative stress/attack. If high-intensity exercise causes greater oxidative attack, shouldn't the antioxidant system have greater adaptation? Include experimental data on ALDH2, antioxidant system and oxidative attack.

What are the practical applications of the data? How to transfer this knowledge to practice?

Calculate the effect size between comparisons.

Author Response

We are grateful to Reviewer #1 for the detailed review and useful comments. As indicated in the responses that follow, we have taken care to address all these comments in the revised version of our paper. We hope that the revisions in our paper and our responses to the comments are satisfactory.

  1. Introduction

A larger physiological argument must be explored to justify a possible effect of polymorphism on muscle oxidative attack. The physiological link to the construction of the hypothesis around the polymorphism is the main limitation of the study.

Thank you for your comments. We agree with the comments. However, unfortunately, we could not address the specific mechanism of the more detailed physiological basis for the effect of genetic polymorphisms on oxidative stress in skeletal muscle. Therefore, we acknowledge that the study's main limitation is the physiological link to constructing the hypothesis around the polymorphism.

What is the genotypic frequency in other populations of GG, AA, and GA?

we added the sentence in results session. Frequency is similar to those reported for other Japanese population groups.

Other ethnicity, such as European, there is about 0% of Minor allele frequency.

  1. Method

Inform the number of the ethics committee.

We added the number of ethics in Methods 2.1.

Include the criteria for classifying the international, national and regional levels of athletes. I suggest that a comparison be made between the levels of the athletes (international, regional, and national). 

We have added the information of criteria for athletic status in Methods 2.2. In addition, we have investigate the association between the athletic status and genotype frequency in Tables 2 and 3.

It is unclear how the absolute and relative Total Record by body mass was calculated.

We added the information of the absolute and relative total records by body mass in Methods 2.2.

Inform if the best results were self-declared, analyzed in loco or consulted in an official record.

We added the information about this suggestion. As for the best record, each athlete was asked to respond to a questionnaire with his/her total personal best record in an official competition (classic powerlifting or weightlifting competition).

General

Avoid the terms: "significant difference", if it was described that there was a difference, it does not need to be redundant to say that it was "significant" or "statistical"

 We revised the word "significant" if it was described that there was a difference.

The rationale for ALDH2 revolves around oxidative stress. Consequently, in older people, oxidative damage over the years shows greater cumulative damage. The average age of the powerlifters group was almost 10 years older than the other group. What do you think about it?

Thank you for an important suggestion. In competitive athletes, the effects of age-related oxidative stress may be a negative factor in achieving high performance. However, we don’t have a data of the association between age-related oxidative stress and performance. Therefore, this effect may not significantly affect the oxidative damage in powerlifters. We added this part to the discussion.

Why wasn't 3 groups analyzed: GG, AA and GA? Just GG vs AA+GA? The sample n seems to me enough to have statistical power in the subdivision into 3 groups.

Thank you for your comments. We added three genotypes (GG, GA, and AA) in the result section. we agree that information on three genotypes should be included to show the relationship between polymorphism and performance comprehensively.

What are the limitations of the study?

 As a major limitation, we failed to consider factors, such as alcohol consumption habits and the daily training status of each athlete, which probably influence athletic conditions and performance. Moreover, all athletes could not achieve their best performance because of different athletic experiences and other reasons. Therefore, longitudinal studies are required. Moreover, we should clarify the effect of other genetic polymorphisms.

As in the Introduction, the physiological mechanism that explains the influence of polymorphisms on mitochondrial and muscle oxidative stress needs to be better connected.

 Thank you for your suggestion. we agree with your comment, but evidence of ALDH2 and muscle phenotype is limited and it is unknown in human muscle. we think further investigation is necessary because many aspects of the ALDH2rs671 polymorphism cannot be still clearly explained in terms of the mechanism of oxidative stress levels in mitochondria and muscles. Therefore, we could not mention it this time.

The endogenous antioxidant system is the most important in combating reactive species. The adaptation of the antioxidant system depends on oxidative stress/attack. If high-intensity exercise causes greater oxidative attack, shouldn't the antioxidant system have greater adaptation? Include experimental data on ALDH2, antioxidant system and oxidative attack.

We agree that the adaptation of the antioxidant system depends on oxidative stress/attack. However, unfortunately, we do not have experimental data on ALDH2, antioxidant system, and oxidative attack to support these hypotheses.

What are the practical applications of the data? How to transfer this knowledge to practice?

Thank you for an important question. The ALDH2 genotype might be provide useful information (e.g., anti-oxidative supplement and genotype-based customization for conditioning) for athletes, especially in strength-related athletes and their strength and conditioning coaches and sports coaches.

Calculate the effect size between comparisons.

We added the effect size (partial η2) to the result part.

Reviewer 2 Report

This is a well organised manuscript and I applauded authors for the concept and methodology of the study. My comments are feedback that I have gotten myself on my own publications and experiences I have had. I hope that these comments will help enhance the current manuscript so that it can be the best version possible, rather than diminish the value of it.

I have downloaded the manuscript as a PDF and notices that is it already in publication form. Edited form and line numbers would make it much easier to give specific feedback. However the feedback is as followed:

Major amendments:

1.       Introduction first paragraph. As a physiologist is it important to differentiate between power and strength? Often in the exercise science literature these two terms have been incorrectly used on multiple occasions. It is fundamental as we move forward that these two words are not used interchangeable. Might be worth changing to ‘power and strength athletes’ as these are different in their respective training and physiology. Alongside this a definition of both and how they are different is necessary for the readers. Especially since you have split them into separate groups.    

In a recent meta-analysis Chung et al., 2021 has shown that strength and power phenotypes respond differently and play different roles in genetic variability. Reference if you deem it applicable.  https://journals.plos.org/plosone/article?id=10.1371/journal.pone.0249501

2.       In paragraph two, you state the alcohol flush reaction (red glow). From reading this I felt that this section was dismissed fairly early. Could this be elaborated on in terms of physiology / biology, why this happens, what this means (i.e. Asians alcohol metabolism is hindered) and its effects on east Asians in terms of exercise / ability, then this could be supported by your previous study.

3.       Methods paragraph one. What was the rational / criteria to who was an “athlete” and who was a control?

4.       Methods 2.2. I think it is fundamental either here or somewhere in the introduction (as stated in point 1), that you give some background on the differences between powerlifters and weightlifters and how you are using Olympic lift protocols to determine the athletic performance as this is not clear from the two sentences you have given in this short section.

5.       Methods 2.3. could more information be given to the protocol of the genetic analysis? When was this collected, how was this collected, how were the samples stored? Did participants provide the salvia samples in a laboratory or were kits sent out to be collected at home etc?

6.       It is briefly touch upon, but I think this study would benefit massively to expand on the limitations / future directions section if possible. One massive limitation is that the study does not have any information on experience level or how trained the athletes are, which could constitute to the wide variance of the results and performance between individuals when comparing groups. Age has been mentioned but would also be good to mention the differences in sex (I know this was mentioned in the results section but could elaborate within the discussion) and how these might change. Other genetics and biological factors play a massive role in things such as testosterone, Hypertrophy, and CSA which is different in males and females, as you have included both and adjusted for sex in your results I would expect this to be mentioned in the discussion.

Minor amendments:

7.       Discussion section: “Kobayashi et al. showed a significantly smaller muscle fiber diameter in ALDH2 KO mice compared to that of wild-type mice, suggesting that the reduced muscle weight observed in mice with ALDH2 deficiency was a consequence of muscle atrophy [23].” Within the manuscript you have used referencing either in place of names or right after the names have been used. Can we keep this consistent, and reference / citation be added right after the use of Kobayashi et al.

8.       Discussion section: “smaller muscle fiber diameter in ALDH2 KO mice” I know KO is commonly used as “Knockout” but this has not been defined anywhere in the test could this please be included.

9.       In the discussion paragraph 3 and 4 there is a lot of research on mice and how this might transition into humans. Could these statements be made more clear that “research show s this happens in mice, therefore there is strong ground to believe that this could also be the case in humans …. Our research agrees with this because ….”. (i.e. “Therefore, we suggest that the A allele of the ALDH2 rs671 polymorphism adversely affects the muscle strength phenotype by impairing muscle configuration and function in competitive HUMAN athletes based on research in mice and the observations within this study.”).

Author Response

We are grateful to Reviewer # 2 for the detailed review and useful comments. As indicated in the responses that follow, we have taken care to address all these comments in the revised version of our paper. We hope that the revisions in our paper and our responses to the comments are satisfactory.

This is a well organised manuscript and I applauded authors for the concept and methodology of the study. My comments are feedback that I have gotten myself on my own publications and experiences I have had. I hope that these comments will help enhance the current manuscript so that it can be the best version possible, rather than diminish the value of it.

I have downloaded the manuscript as a PDF and notices that is it already in publication form. Edited form and line numbers would make it much easier to give specific feedback. However the feedback is as followed:

Major amendments:

  1. Introduction first paragraph. As a physiologist is it important to differentiate between power and strength? Often in the exercise science literature these two terms have been incorrectly used on multiple occasions. It is fundamental as we move forward that these two words are not used interchangeable. Might be worth changing to ‘power and strength athletes’ a s these are different in their respective training and physiology. Alongside this a definition of both and how they are different is necessary for the readers. Especially since you have split them into separate groups.    

In a recent meta-analysis Chung et al., 2021 has shown that strength and power phenotypes respond differently and play different roles in genetic variability. Reference if you deem it applicable.  https://journals.plos.org/plosone/article?id=10.1371/journal.pone.0249501

Thank you for your good suggestion. We agree that it is essential to distinguish between power athletes and strength athletes since these are different in their respective training and physiology. We added the importance of distinguishing power-oriented and strength-oriented athletes when examining the association between phenotypes and polymorphisms in the discussion section, citing the paper you shared.

  1. In paragraph two, you state the alcohol flush reaction (red glow). From reading this I felt that this section was dismissed fairly early. Could this be elaborated on in terms of physiology / biology, why this happens, what this means (i.e. Asians alcohol metabolism is hindered) and its effects on east Asians in terms of exercise / ability, then this could be supported by your previous study.

Thank you for your comment. We had revised this sentence.

  1. Methods paragraph one. What was the rational / criteria to who was an “athlete” and who was a control?

In Method 2.2, we added criteria for athletes and controls in present study.

  1. Methods 2.2. I think it is fundamental either here or somewhere in the introduction (as stated in point 1), that you give some background on the differences between powerlifters and weightlifters and how you are using Olympic lift protocols to determine the athletic performance as this is not clear from the two sentences you have given in this short section. Differences between powerlifting and weightlifting disciplines have been added. We have also included a description of how performance is determined in these disciplines.

In Methods 2.2., We added differences between powerlifting and weightlifting disciplines. We have also included a description of how performances are determined in these events.

  1. Methods 2.3. could more information be given to the protocol of the genetic analysis? When was this collected, how was this collected, how were the samples stored? Did participants provide the salvia samples in a laboratory or were kits sent out to be collected at home etc?

   We added more information on how the samples were collected in method 2.3.

  1. It is briefly touch upon, but I think this study would benefit massively to expand on the limitations / future directions section if possible. One massive limitation is that the study does not have any information on experience level or how trained the athletes are, which could constitute to the wide variance of the results and performance between individuals when comparing groups. Age has been mentioned but would also be good to mention the differences in sex (I know this was mentioned in the results section but could elaborate within the discussion) and how these might change. Other genetics and biological factors play a massive role in things such as testosterone, Hypertrophy, and CSA which is different in males and females, as you have included both and adjusted for sex in your results I would expect this to be mentioned in the discussion.

   In the discussion sections, we added sex-related factors, such as testosterone, which potentially impact performance.

Minor amendments:

  1. Discussion section: “Kobayashi et al. showed a significantly smaller muscle fiber diameter in ALDH2 KO mice compared to that of wild-type mice, suggesting that the reduced muscle weight observed in mice with ALDH2 deficiency was a consequence of muscle atrophy [23].” Within the manuscript you have used referencing either in place of names or right after the names have been used. Can we keep this consistent, and reference / citation be added right after the use of Kobayashi et al.

    We changed the position of reference right after Kobayashi et al.

  1. Discussion section: “smaller muscle fiber diameter in ALDH2 KO mice” I know KO is commonly used as “Knockout” but this has not been defined anywhere in the test could this please be included.

   Thanks for pointing it out. We incorrectly listed the citation as KO mice, not Transgenic mice.

  1. In the discussion paragraph 3 and 4 there is a lot of research on mice and how this might transition into humans. Could these statements be made more clear that “research show s this happens in mice, therefore there is strong ground to believe that this could also be the case in humans …. Our research agrees with this because ….”. (i.e. “Therefore, we suggest that the A allele of the ALDH2 rs671 polymorphism adversely affects the muscle strength phenotype by impairing muscle configuration and function in competitive HUMAN athletes based on research in mice and the observations within this study.”).

     We have rewritten this part to explain it more clearly in line with the comments.

Round 2

Reviewer 1 Report

I have no more consideration